# Intravenous Administration of Coenzyme Q10 in Acute Period of Cerebral Ischemia Decreases Mortality by Reducing Brain Necrosis and Limiting Its Increase within 4 Days in Rat Stroke Model

**DOI:** 10.3390/antiox9121240

**Published:** 2020-12-07

**Authors:** Obolenskaia Olga Nikolaevna, Gorodetskaya Evgeniya Aronovna, Kalenikova Elena Igorevna, Belousova Margarita Alekseevna, Gulyaev Mikhail Vladimirovich, Makarov Valery Gennadievich, Pirogov Yury Andreevich, Medvedev Oleg Stephanovich

**Affiliations:** 1Faculty of Medicine, Lomonosov Moscow State University, 119991 Moscow, Russia; obolenskaia@fbm.msu.ru (O.O.N.); Gorodetskaya@fbm.msu.ru (G.E.A.); eikaleni@fbm.msu.ru (K.E.I.); gulyaev@physics.msu.ru (G.M.V.); medvedev@fbm.msu.ru (M.O.S.); 2Research and Production Association «House of Pharmacy», 188663 Saint Petersburg, Russia; yupi@physics.msu.ru; 3Faculty of Physics, Lomonosov Moscow State University, 119991 Moscow, Russia; info@doclinika.ru

**Keywords:** coenzyme Q10, ubiquinol, ubiquinone, brain ischemia, neuroprotection, stroke, antioxidant

## Abstract

Oxidative stress plays a key role in the pathogenesis of ischemic stroke. Coenzyme Q10 has a multi-targeting effect and may protect the brain against ischemic damage. The aim of our study was to evaluate the neuroprotective potential of ubiquinol by its intravenous administration. The study was performed on rats; a stroke was modeled by occlusion of the middle cerebral artery. On days 1 and 4 after ischemia, the neurological deficit and volume of the brain lesion were determined by MRI and TTC staining. Intravenous administration of coenzyme Q10 led to a decrease in rat mortality rate, improvement in neurological status, and decrease in the brain necrosis area in acute and delayed period after cerebral ischemia. A single intravenous administration of ubiquinol led to a limitation of the size of the brain damage for at least four days after ischemia. Thus, intravenous administration of coenzyme Q10 has a persistent neuroprotective potential. This finding suggests a possible therapeutic role of ubiquinol in acute ischemic conditions.

## 1. Introduction

The problem of ischemic brain damage is acute in modern healthcare. The death rate from stroke continues to be the leading contributor to the general mortality rate in the population. In addition, ischemic stroke carries important socio-economic consequences, leading to disability in working-age people [1].

Despite the considerable advances in modern pharmacology, the only effective therapy for ischemic stroke is thrombolytic therapy, which, unfortunately, can only be carried out in 2–5% of patients, due to contraindications and a narrow time window in which it will be effective and safe [2]. Thrombectomy is also an effective method of stroke treatment, which allows to restore blood flow to the ischemic brain tissue, reduces patient mortality, and improves neurological outcomes. The search for new substances is urgent. Neuroprotective agents are capable of interrupting a cascade of pathophysological reactions and protecting the brain from the damaging effects of ischemia. Timely restoration of cerebral blood flow limits the area of necrosis, but on the other hand, reperfusion injury contributes further to brain damage, which is called ischemic-reperfusion syndrome. The resumption of blood flow leads to the activation of oxidative stress: it initiates the formation of free radicals and reactive oxygen species, activates lipid peroxidation, DNA damage, triggers apoptosis cascades, and develops neuronal death. Under condition of oxidative stress, the activation of endogenous antioxidant systems is insufficient to prevent massive neuronal death [3].

Coenzyme Q10 is present in all cells of the body and has many functions: as an antioxidant, prevents the opening of mitochondrial pores and the destruction of mitochondria, has an anti-apoptotic effect, improves endothelial function, and has a complex anti-inflammatory effect [4]. Such a large number of application points for the action of Coenzyme Q10 makes it an attractive and promising substance for the treatment of stroke, both in combination with thrombolytic therapy or thrombectomy, and as an independent remedy [5]. Now it is used in the complex therapy of heart failure, diabetes, and atherosclerosis. There are clinical studies that have shown the effectiveness of coenzyme Q10 in such neurodegenerative diseases as Parkinson’s disease, Alzheimer’s disease and Huntington’s chorea, and a number of other diseases. However, in these studies, coenzyme Q10 was used orally for a long time in a high dosage, which is due to the low bioavailability when taking orally and the lack of forms for parenteral use at the time of the study. Since stroke is an urgent condition, intravenous drug administration is necessary [6].

In the current work, the neuroprotective effect of a new innovative form of coenzyme Q10 for intravenous administration was studied. This form was developed by scientists at the House of Pharmacy in St. Petersburg, Russia, and is a safe, stable, and sterile 1% solution of ubiquinol. It is well known that coenzyme Q10 can exist in two forms—ubiquinol (reduced form) and ubiquinone (oxidized form). It is ubiquinol that exhibits the greatest antioxidant activity; however, this molecule is less stable and a transition from one form to another occurs in tissues [7,8]. This study compared two dosage forms of coenzyme Q10 administered to stroke animals under equal conditions.

In addition, in our experiment, the dynamics in the increase of brain damage zone was studied during four days after the onset of ischemia. There is no consensus among scientists regarding the increase in the zone of brain necrosis in the delayed period after ischemia. Some researchers point to the growth of the lesion focus due to the penumbra zone and traumatic effect of edema [9]. Others argue that the main zone of necrosis is formed during the first day, and then involution of the damaged zone occurs with the formation of a glial scar [10]. The aim of the second part of our study was to study the dynamics of the brain damage zone in rats using MRI (magnetic resonance imaging) over four days and to assess the possibility of limiting this zone using a single intravenous administration of ubiquinol in the acute period of stroke.

## 2. Materials and Methods

### 2.1. Experimental Groups and Study Design

All experimental procedures were performed in accordance with the Guide of the Care and Use of Laboratory Animals, 8th edition by National Research Council, and were approved by Local Animal Care Committee (Project No. 01201353963). The study was performed on adult Male Wistar rats weighing 300–350 g, which were purchased from the Institute of Biomedical Problems (Khimki, Russia) and housed in 12 light/dark cycle with free water and food access for at least 7 days prior the experiment. To evaluate the neuroprotective efficacy of Ubiquinol and Ubiquinone, we performed two independent experiments. In the first one, we estimated the neuroprotection ability of Ubiquinone and Ubiquinol in the acute period of cerebral ischemia. Rats were subjected to 60 min of focal cerebral ischemia by middle cerebral artery occlusion (MCAO), a model design as below. Substances were administered 45 min after the onset of ischemia (15 min prior to reperfusion). In the first experiment 24 h after ischemia, the rats were tested for neurological deficit, then decapitated; brain infarct size and coenzyme Q10 brain levels were measured. All animals were randomly divided into 6 groups: MCAO+NaCl, n = 21; MCAO+Vehicle, n = 17; MCAO+UL (Ubiquinol), n = 12; MCAO+UN (Ubiquinone), n = 12; Sham-operated, n = 10; Native, n = 4.

In the second experiment, we evaluated the neuroprotective efficacy of ubiquinol in delayed period. We measured the dynamic of necrotic brain area size for 4 days. Rats were subjected to 60 min focal cerebral ischemia; substances were administered 15 min prior to reperfusion. The brain infarct volume was measured by MRI at 24 h and 4 days after ischemia in each rat. Due to the high mortality rate of animals, only surviving animals were included in the final selection of groups. Rats were divided into 2 groups: MCAO+Veh, n = 25; MCAO+UL (Ubiquinol), n = 5.

### 2.2. Investigational Substances

Among the studied substances, the main one was 1% solution of ubiquinol. It is an innovative water-soluble form of ubiquinol for parenteral administration. It is sterile, safe, and stable. It is manufactured by the research and production association “House of Pharmacy” St. Petersburg, Russia (patent # RU2635993-C1).

Composition (1 mL):Main active compound: Ubidecarenol 10 mg;Excipients: Macrogol glycerol ricyl oleate 80 mg,Polysorbate 40 mg,Ascorbic acid 1 mg,Sodium EDTA 0.5 mg,Sodium chloride 0.9% ad 1 mL.

As a reference substance, 1% Ubiquinone of the same production and composition was used, which contained a Ubidecarenone (10 mg/mL) instead of Ubidecarenol (10 mg/mL).

For a group of animals without the introduction of active substances, a vehicle was used, which contained only excipients in the composition listed above, or a saline (0.9% sodium chloride solution).

### 2.3. Substances Administration

All substances were administered intravenously via PE catheter, introduced into the left femoral vein. All substances were kindly provided by the House of Pharmacy, Saint Petersburg, Russia. We used 1% Ubiquinone (UN) solution, 1% Ubiquinol (UL) solution, and standard vehicle solution (placebo). Both active substances were injected in a dose 30 mg/kg 45 min after the onset of ischemia. Saline-treated, vehicle-treated, and sham-operated rat received saline/vehicle in an equal volume (about 1, 2 mL).

### 2.4. Focal Cerebral Ischemia Model

Focal cerebral ischemia was performed by the Longa EZ method [11]. Rats were anesthetized by a single intraperitoneal injection of chloral hydrate (300 mg/kg). Atropine sulfate was administered subcutaneously to prevent bradycardia and bronchial hypersecretion. Rats were placed in a supine position; incision was made on the left of the midline. Novocain was injected subcutaneously in the projection of surgical incision to prevent pain. The left common carotid artery (CCA), internal carotid artery (ICA), and external carotid artery (ECA) were exposed. Two microsurgical clips were placed on CCA and ICA. Branches of ECA—occipital artery and superior thyroid artery were cuted by bipolar coagulation. On the ECA, an initial partial arteriotomy (3 mm) was performed and a 3.0 cm 4–0 monofilament nylon filament with a silicon-coated tip was introduced into the ICA via ECA through CCA bifurcation. The filament was gently advanced into ICA until slight resistance was felt and the filament blocked the origin of middle cerebral artery. The filament (Doccol, single-use silicon coat filament) was left for 60 min to perform a strong ischemia, then it was slowly removed to perform reperfusion. The incision was closed with a suture and rats were placed in individual cages with free access to food and water to recover from the operation. All sham-operated rats were subjected to the same surgical manipulations, except filament introduction.

### 2.5. Neurological Function Measurement

To evaluate the neurological deficit, we used the accepted modified neurological severity score (mNSS) proposed by Reglodi et al. [12]. This scale provides a behavior deficit score after reviewing motor (4 test) and sensor (2 test) functions. Motor tests are: 1—spontaneous activity in the cage (5 min detection); 2—symmetry of limb movements during spontaneous activity; 3—symmetry of limbs when hanging a rat by the tail; 4—ability to climb the vertical grid. Sensor tests determine sensitivity when touching the sides of the rat’s body and vibrissae (test 5 and 6, respectively). Each test ranged from 0 to 3, the total score was 18 (the lower the total score, the greater the neurological deficit). Neurological tests were performed before decapitation 24 h and 4 days post-surgery. mNSS was assessed by a well-trained investigator who was blinded to the groups.

### 2.6. Infarct Volume

Rats were euthanized with an overdose of anesthesia in compliance with ethical standards. After the decapitation brain was removed, it was frozen at −20 °C for 15 min, then gently cut into 5 coronal slices of 2 mm thickness each. Slices were stained with metabolic dye 2% TTC (2,3,5 triphenyltetrazolium chloride) for 30 min at 37 °C, then photographed using a digital camera.

In the second experiment, we measured infarct size dynamics using MRI pictures. 24 h and 4 days after ischemia, each rat was put under slight anesthesia using inhalation of 1.5% Isoflurane and MRI scans were performed. Studies were performed on a 7.05 T Bruker BioSpec 70/30 USR MR scanner driven by a ParaVision^®^ 5.1 console and equipped with a 105 mT/m gradient amplitude device. T2-weighted MR images were obtained in axial projection using spin-echo pulse sequence RARE (rapid acquisition with relaxation enhancement) with the following scan parameters: field of view: 2.56 × 2.56 cm; matrix: 156 × 156; bandwidth: 25,000 Hz; TR: 5500 ms; TE: 14.04 ms; TEeff: 56.20 ms; RARE factor: 8; number of slices: 20; slice thickness: 0.8 mm; number of averages: 2; total scan time: 3 min 29 s. Images were analyzed using ImageJ soft (Wayne Rasband, Research Services Branch of the National Institute of Mental Health).

Brain infarct volume was presented in % to ipsilateral hemisphere volume, and calculated as: S_ish_ = S_cont_ − (S_ipsi_ − S_necr_), V_ish_ = ΣS_ish_ × d (S_ish_—area of ischemia; S_cont_—area of contralateral hemisphere; S_ipsi_—area of ipsilateral hemisphere; S_necr_—area of necrosis; V_isch_—volume os ischemia; d—brain slice thickness 2 mm).

### 2.7. Tissue Preparation and HPLC Analysis

After brain staining in TTC necrotic area from each slice was removed, the brain was separated in two parts—ipsilateral and contralateral hemisphere and was frozen under −70 °C until further measurement. Then brain samples were unfrozen, homogenized in 96% C2H5OH (ratio 1:4) by ultrasound homogenizer. 100 μL oh homogenate was transferred in an individual eppendorf for further extraction. N-Hexan (250 μL) was added to the samples and were shaken for 10 min, then centrifugated at 5000× *g* for 5 min. The upper hexan layer was collected and the extraction was repeated. The total extract amount was evaporated and dissolved in 100 μL of 96% ethanol. CoQ10 levels were measured using high-performance liquid chromatography with electrochemical detection (Environmental Sciences Associate, Inc., Chelmsfort, Massachusetts, USA): model 580 pump and electrochemical detector “Coulochem II”, in isocratic mode on Luna column 150 × 4.6 mm with sorbent C18 (5 μm) at a flow rate of the eluent of 1.3 mL/min.

### 2.8. Statistical Processing Methods

The data are presented as Mean ± Standard Deviation. Between-groups comparisons were performed using the non-parametric Mann-Whitney U-test (Soft Statistic 8.0). To compare repeated measurements within the group, the nonparametric Wilcoxon test was used. A *p* value of <0.05 was considered statistically significant.

## 3. Results

### 3.1. Mortality

The middle cerebral artery occlusion model is a rather traumatic model of stroke and leads to the formation of a large lesion area in the middle cerebral artery basin. Animals that died during the operation or within 2 h after coming out of anesthesia were not included in the final mortality calculation (their deaths were taken as complications of the operation).

In the first experiment, the death rate in the saline (MCAO+NaCL) and vehicle (MCAO+Veh) groups was the same and amounted to 43% (9 of 21 animals died) and 41% (7 of 17 died), respectively. In the sham-operated group, the occlusion of the common carotid artery for 60 min did not lead to cerebral ischemia due to collateral blood flow, and no deaths were recorded in this group. In the ubiquinone-treated group (MCAO+UN), the mortality rate was minimal (8%: 1 of 12 animals died) and was significantly lower compared to the MCAO+NaCL and MCAO+Veh groups, *p* < 0.01. Intravenous administration of ubiquinol (MCAO+UL) increased the survival rate up to 100%—not one of the 12 operated rats died by the end of 1 day after occlusion of the middle cerebral artery.

In the second experiment lasting 4 days, the mortality of animals increased compared to 1 day because of the edema and expansion of brain damage. In the MCAO+Veh group, the mortality was 80% (20 of 25 animals died). In the ubiquinol-treated group (MCAO+UL), the mortality was significantly lower—20% (1 in 5 animals died), *p* < 0.01.

### 3.2. Neurological Outcomes

Local cerebral ischemia led to the formation of neurological deficits. When registering on the mNSS scale, mainly motor deficits was detected (tests 1–4), but sensor deficits (test 5,6) was also recorded. No neurological deficit was observed in the sham-operated animals; the average score was 17.7 ± 1.3. The total score in the MCAO+NaCL and MCAO+Veh groups was significantly decreased (*p* < 0.01) and was 7.4 ± 2.3 and 7.1 ± 1.6, respectively. In the group with intravenous administration of ubiquinone and ubiquinol, the neurological status of the animals was significantly higher than in the vehicle-treated group and amounted to 10.6 ± 2.1 and 11.2 ± 1.2, respectively, *p* < 0.05. However, there was no significant differences between MCAO+UL and MCAO+UN groups, *p* = 0.4 (Figure 1).

In the second experiment, it was shown that within four days, the physical condition of animals in the MCAO+Veh group worsened and neurological deficit increased. 24 h post-surgery, the total neurological score was 8.6 ± 1.9, and by the fourth day, a significant decrease in the score to 5.6 ± 0.5 was noted, *p* < 0.05. It was mainly the motor functions that suffered; some of the animals were alive, but showed minimal motor activity and performed only slow rotational movements around the paralyzed side. In the MCAO+UL group, the total score was higher on the first day (11.0 ± 0.7) compared with the vehicle group, *p* < 0.05. However, for four days after the operation, the neurological status was not impaired in the treated group. It is likely that a single intravenous injection of ubiquinol was sufficient to prevent deterioration in neurological status. The overall neurological score was 11.2 ± 0.4 (Figure 2).

### 3.3. Brain Infarct Volume

Focal cerebral ischemia led to the formation of a necrosis field surrounded by a zone of penumbra-ischemic area, where neurons are in a reversible hypoxia. With the metabolic dye TTC, it was possible to stain healthy viable brain tissue, while the necrosis area remained white.

In the first acute experiment, the zone of necrosis (in relation to the ipsilateral hemisphere) in the MCAO+Veh group (31.3 ± 9%) was comparable to the MCAO+NaCL group (32.2 ± 13.4%), *p* = 0.9. Intravenous administration of 1% ubiquinol and ubiquinone solutions increased the resistance of brain tissue to ischemia and led to the limitation of the necrosis zone. In the MCAO+UL group, the necrosis zone was 18.7 ± 13.9%, which was significantly lower than in the saline and vehicle groups, *p* < 0.05. In the MCAO+UN group (20.7 ± 9.9%), a decrease in the area of necrosis was also observed, *p* < 0.05. At the same time, no statistically significant differences were found between animals receiving a single intravenous administration of ubiquinone or ubiquinol, *p* = 0.5 (Figure 3).

In the second experiment, the dynamics of the growth of brain lesion was assessed for four days after the occlusion of the middle cerebral artery. When evaluating the MRI images, the total area of brain damage was measured—both the zone of necrosis, which looked whiter on T2 images, and the surrounding area of ischemic penumbra. It should be noted that the mortality of the animals in the vehicle group was very high (80%), and the animals with a large lesion did not survive until the end of the experiment. Thus, the size of the brain damage, measured 24 h post-surgery, in rats that did not survive until the end of the experiment (up to 4 days) was 44.2 ± 10.2%, which was comparable to the data obtained in our first experiment. In the vehicle group, only animals with a small area of brain damage survived until the end of the experiment and were included in the final data processing. Twenty-four hours after the operation, the zone of damage in MCAO+Veh group and MCAO+UL group amounted to 24.4 ± 1.1 and 24.7 ± 5.7%, respectively, *p* = 0.4. However, just 1 animal from the ubiquinol group died. Within four days, in the MCAO+Veh group, an increase in the area of brain lesions was observed almost twice. The zone increased by 96% and amounted to 48.0 ± 9.3%, *p* < 0.05. In the group that received a single intravenous injection of ubiquinol, there was no growth in the affected area; it remained the same size by the end of the first day after ischemia—24.5 ± 5.6% (Figure 4). Thus, the administration of ubiquinol allows to protect brain tissue from the damaging effects of edema and to prevent the spread of damage through the penumbra.

To carry out a comparative analysis of the two methods to assess the affected area, the morphological measurement of the necrosis area was used using TTC staining on the 4th day after occlusion. Obtained data matched with the size of the lesion measured by MRI. Thus, in the MCAO+UL group, the size of the necrosis zone was 22.7 ± 2.5% (when measured on MRI, it was 24.5 ± 5.6%). Most likely within 4 days the effect of cerebral edema and ischemic penumbra becomes not as significant as 24 h post-surgery; therefore, the entire damage zone is represented exactly with brain necrosis. In the MCAO+Veh group, the zone of necrosis measured after TTC staining was 39.6 ± 8.7% and did not significantly differ from the zone obtained by the MRI images—48.0 ± 9.3%, *p* = 0.5. The insignificant difference in size could be explained as such: when analyzing MRI images, both the necrosis zone and the surrounding ischemic penumbra are taken into account, and it is difficult to match the border between these zones by evaluating the scans. When staining TTC, the necrosis zone is clearly visualized as a dead white tissue. Most likely, in rats from the vehicle group, the penumbra continues to exist for at least 4 days, and intravenous administration of ubiquinol leads to a restriction of the necrosis zone and inhibits its development already on the first day after the operation.

The obtained data convincingly indicate that without treatment, the area of brain damage continues to increase in the delayed period after ischemia within at least four days, perhaps because of the penumbra zone growing, which opens up additional opportunities for therapeutic action, for example, repeated administration of neuroprotective substances. According to the data published earlier, it was shown that after a single injection of ubiquinol, elevated coenzyme Q10 levels persisted in the brain (by 60%) and in the liver (20-times) at least for four days [13]. Most likely a single injection of ubiquinol is sufficient to limit brain damage in the delayed period.

### 3.4. CoQ10 Brain Level

The coenzyme Q10 level in the brain was measured separately in the ipsilateral hemisphere (after removal of the necrosis zone) and contralateral hemisphere. In the group of native animals and sham-operated group, there were no significant differences in the tissue level of coenzyme Q10 and the levels were 21.7 ± 1.2 mkg/g and 20.2 ± 1.7 mkg/g, respectively, *p* = 0.7.

In the MCAO+NaCL group, there was a significant decrease in the level of endogenous coenzyme Q10 in both hemispheres. In the ipsilateral hemisphere, which contained the tissue most affected by ischemia, there was a decrease in the level of coenzyme Q10 by 32% compared to native and sham-operated rats—14.8 ± 1.9 mkg/g, *p* < 0.05. In the contralateral hemisphere, a 24% depletion of the endogenous coenzyme Q10 level was also noted and was 16.4 ± 2.3 mkg/g, *p* < 0.05. In the MCAO+Veh group, coenzyme Q10 level was significantly reduced in ipsilateral hemisphere (16.0 ± 1.7 mkg/g) compared not only to native rats and rats that received coenzyme Q10, but also with contralateral hemisphere (19.7 ± 1.4 mkg/g) within their group, *p* < 0.05.

A single intravenous administration of ubiquinol led to the restoration of brain tissue levels of coenzyme Q10 in both hemispheres to the value of native animals. In the ipsilateral hemisphere, the coenzyme Q10 level was 20.4 ± 4.4 mkg/g, and in the contralateral hemisphere −22.6 ± 5.5 mkg/g.

Intravenous administration of ubiquinone also led to the restoration of tissue levels of coenzyme Q10 in both hemispheres to the level of native animals: 25.2 ± 7.5 mkg/g in the contralateral and 22.8 ± 4.3 mkg/g in the ipsilateral hemisphere. A tendency for a greater accumulation of coenzyme Q10 in animals receiving intravenous administration of ubiquinone was not significant, *p* = 0.07 (Figure 5).

In the second experiment, the tissue brain level of coenzyme Q10 was determined in the delayed period of ischemia. In the MCAO+Veh group, a significant decrease in tissue levels of coenzyme Q10 in both hemispheres was noted, both in comparison with native animals and with animals receiving intravenous administration of ubiquinol, *p* < 0.01. The level of coenzyme Q10 in the ipsilateral hemisphere was 10.8 ± 0.7 mkg/g and was significantly lower than in the contralateral hemisphere 13.8 ± 1.0 mkg/g, *p* < 0.05. By the 4th day after ischemia, in the group of animals receiving ubiquinol, there was a tendency of a decrease of coenzyme Q10 brain levels in both hemispheres: in the ipsilateral hemisphere, it was 19.5 ± 4.1 mkg/g, and in the contralateral hemisphere, it was −19.1 ± 3.9 mkg/g. However, this decrease was not significant as compared to the native group, *p* = 0.5 (Figure 6).

Thus, a single intravenous administration of ubiquinol in the acute period of stroke led to a persistent accumulation of coenzyme Q10 in the brain tissue, which most likely made it possible to maintain the level of antioxidant protection against ischemic-reperfusion injury and provided a neuroprotective effect in the delayed period after cerebral ischemia.

## 4. Discussion

Ischemic brain damage causes a lack of oxygen in local tissue and a number of metabolic disturbances in the tissue by a lack of nutrients and the accumulation of toxic decay products. The cascade of ischemic reactions includes the activation of anaerobic glycolysis, development of lactic acidosis, and cytotoxic edema of cells. Aggravation of ischemia leads to a decrease in ATP synthesis, disruption of ion channels, activation of necrosis, and apoptosis [14]. The central area of ischemia, called the nucleus, undergoes irreversible changes very early—within the first 10 min from the onset of ischemia. For 3–6 h, this zone remains surrounded by the ischemic penumbra, in which the tissue is reversibly ischemic. The formation of the main part of cerebral infarction occurs in the first 4–6 h after the onset of ischemia; however, the processes launched in the first hours of ischemia make a significant contribution to distant brain damage. Lactic acidosis, calcium excitotoxicity, oxidative stress, and triggered apoptosis cascades continue to develop up to 24 h and could lead to an increase in the affected area in the long term up to 7 days. Then, the regressive process predominates with the formation of cerebral cysts and glial scar [15].

Despite the fact that the best therapeutic effect in acute cerebral ischemia is a timely restoration of blood flow (with spontaneous recanalization of a thrombi or with fibrinolytics), reperfusion produces an additional contribution to brain damage—this is called reperfusion syndrome or oxygen paradox. Under conditions of hypoxia, the cells accumulate many reduced components of the mitochondrial respiratory chain. When the flow of oxygen into these zones is resumed, electrons are discharged bypassing the respiratory chain and oxidative stress is induced—many highly active free radicals are formed. The increase in brain damage area occurs precisely due to the penumbra. Ischemic penumbra is the target for neuroprotective drugs, since it contains tissue with functional but potentially reversible damage [16].

According to the multi-targeting effect of coenzyme Q10, it is a promising agent in neuroprotection. It works as a free radical scavenger, reduces oxidative stress, has anti-apoptotic effects and stabilizes mitochondrial pores, and reduces calcium overload and excitotoxicity. Studies on coenzyme Q10 as a neuroprotector began long ago and continues to this day [17]. There is a large number of preclinical and a number of clinical studies that have shown a certain effectiveness of coenzyme Q10 when taken orally for a long time in neurodegenerative diseases such as Alzheimer’s disease, Parkinson’s disease, amyotrophic lateral sclerosis, and some others [17]. A key problem to the use of coenzyme Q10 in acute conditions such as ischemia is its low oral bioavailability and high hydrophobicity—currently, the number of dosage forms of coenzyme Q10 for intravenous administration is extremely limited. In a recent published paper, the authors managed to create a water-soluble form of coenzyme Q10 using the antifungal caspofungin [18].

In 2000, a number of papers were published describing the use of coenzyme Q10 when administered orally or intraperitoneally in different models of ischemic brain damage. Studies have shown the ability of coenzyme Q10 to reduce the concentration of toxic products in the brain (MDA), increase the activity of endogenous antioxidant systems, and reduce the mortality rate of rats [19]. Some studies have been published showing the effectiveness of coenzyme Q10 on cultures of neuronal cells subjected to hypoxia: coenzyme Q10 reduced the severity of apoptosis and increased neuronal survival [20].

Earlier, we studied the neuroprotective effect of coenzyme Q10 in the composition of the liquid dosage form for oral administration «Kudesan» [21]. This form is recommended for systematic use in heart failure complex treatment. «Kudesan» is a water solution of solubilized coenzyme Q10 (30 mg/mL), which contains, in addition to coenzyme Q10, alpha tocopherol (4.5 mg/mL), ascorbil palmitat (1 mg/mL), and citric acid (1.6 mg/mL). In the experiment, we administered «Kudesan» intravenously, the safety and good tolerance of the substance was demonstrated, and no toxic effects was found. It was shown that a single administration of «Kudesan» led to a decrease in rat mortality, an improvement in neurological status, and a limitation of the brain necrosis zone on days 1 and 7 [21].

It is known that it is the reduced coenzyme Q10, ubiquinol, that exhibits antioxidant activity. Although there is a balance between the oxidized and reduced coenzyme Q10 in the body in each tissue, it is ubiquinol that is most effective in reducing oxidative stress. In our current study, a new innovative form of ubiquinol for parenteral administration is the main target for study. It is a stable, sterile form of 1% ubiquinol that does not contain alpha tocopherol, which potentiates the antioxidant activity of coenzyme Q10. Thus, the entire effect of the substance administration will be due exclusively to the action of the active substance—ubiquinol, as the vehicle had no neuroprotective effect. We used 1% solution of Ubiquinone of the same manufacturer and a solution of a standard solvent as the reference medicine.

In the current experiment, the mortality rate of animals that did not receive active substances on the first day averaged 42%. Intravenous administration of ubiquinol and ubiquinone solutions led to an increase in the survival rate of animals. The data are consistent with the results that we obtained earlier in the experiment with «Kudesan». Thus, in the previous experiment, the mortality in the saline group was about 33%, and the mortality rate of animals receiving intravenous injection of «Kudesan» was significantly lower 14% (2 out of 14 animals died).

Since in the first experiment, no significant difference was recorded in the groups of animals receiving the vehicle and saline, therefore, for the second experiment, it was decided to leave only the group with the intravenous administration of the vehicle in order to minimize the amount of affected animals and comply with the ethical protocol of the experiment. In the second experiment, the mortality rate of animals in the vehicle group increased twice for four days up to 80%. In the ubiquinol group, animal mortality remained at a low level; only 1 animal did not survive to the end of four days. Thus, a strong ability of ubiquinol to increase animal survival was shown.

The neurological deficit in the group of animals that received saline and vehicle was significantly higher than in the animals treated with coenzyme Q10 in both experiments. In addition, in animals from the vehicle group, significant deterioration in neurological status was noted by the 4th day: a decrease in the total score from 8.6 to 5.6 points disappeared. Some animals could not perform any neurological tests—they were motionless in the corner of the cage and did not respond to touch, or made rotational movements around the paralyzed body side; some had epileptomorphic seizures. Moreover, most of the animals with severe neurological symptoms did not survive until the end of the experiment and were not included in the final data processing. Among the animals receiving intravenous administration of ubiquinol, an increase in the average total score by 1.5 times was noted compared with the vehicle group. Moreover, the neurological status did not deteriorate for four days (Figure 2). The animals remained active and were able to perform tail hanging test and vertical grid climbing test, although they showed muscle weakness in the limbs on the affected side of the body. However, the general physical condition of the animals could be assessed as satisfactory, they did not refuse to eat, and showed tolerable physical activity. At the same time, in our study, a correlation between the size of the brain necrosis zone and the severity of neurological deficit was noted: the greater the brain damage, the lower the overall neurological status score was. In general, this pattern is logical and coincides with the data obtained by other authors on a similar model of cerebral infarction (Figure 7).

In addition, the severity of the neurological deficit correlated with the content of coenzyme Q10 in the brain. A reduced amount of coenzyme Q10 in the brain, especially in the ipsilateral hemisphere, in which the brain centers responsible for the performance of motor functions and sensitivity were located on the opposite side of the body (since the intersection of nerve pathways anatomically lies below the infarction area in the brain) was directly related to the severity of neurological violations—the more coenzyme Q10 decreased, the more severe the deficiency was (Figure 8).

Thus, the improvement in neurological status in groups receiving intravenous injection of ubiquinol or ubiquinone was associated with an increase in the content of coenzyme Q10 in the ipsilateral hemisphere of the brain, which led to the limitation of the area of brain damage.

Acute experiments lasting 24 h have shown the ability of ubiquinone and ubiquinol to limit the area of brain necrosis. The 24-h time period is widely used in assessing the neuroprotective efficacy of medicine and is recommended by most authors for assessing the outcome of acute stroke. In this experiment, both active substances showed similar efficacy—intravenous administration of ubiquinol led to a limitation of the necrosis zone by 40.6%, compared to the group receiving the solvent, and in the group of animals receiving ubiquinone, a decrease in the necrosis zone by 37.5% was noted compared to the animals receiving the solvent (Figure 3; Figure 9).

Since we have previously shown the effectiveness of intravenous administration of a solution of ubiquinone as part of nutraceuticals «Kudesan», in this study ubiquinone was taken as a comparison drug. Some researchers suggest that ubiquinol has an advantage over ubiquinone in terms of the severity of antioxidant activity. In our study, the tendency of ubiquinol to strongly limit the brain necrosis area was shown, but these differences were not statistically significant. As mentioned earlier, there was a correlation between the size of the necrosis zone and the severity of neurological disorders in all experimental groups (Figure 7). In addition, a correlation was noted between the zone of necrosis and tissue content of coenzyme Q10—the greater the content of CoQ10 in the ipsilateral hemisphere was, the smaller the area of brain damage was (Figure 10).

In the current experiment, the most interesting for us was the study of the dynamics of the brain damage zone in the long-term period after the formation of a cerebral infarction. In most studies, the authors directly study the acute period of ischemia (the average duration of the experiment is 24 h); however, of great interest is also the changes that occur in the zone of necrosis during the delayed period. Thus, it is known that a cascade of pathogenetic reactions will grow after the first day of ischemia—cerebral edema continues to develop, the process of calcium excitotoxicity leads to the spread of the necrosis zone through the penumba region further along the brain, and apoptosis increases. A number of authors show that after the first day, there is a further decrease in the zone of necrosis in the brain [10]. It should be noted that such a decrease may be due to the formation of a glial scar, which, as it were, “tightens” the brain tissue, as well as a number of methodological features when calculating the size of the necrosis zone for 1 day. Thus, in our study, we used a formula that takes into account the contribution of cerebral edema to the size of the affected area: in the first 24 h, due to pronounced edema, the brain is hyperhydrated and sometimes the ipsilateral hemisphere may even exceed the contralateral one in size. This must be taken into account in the final calculation of the brain damage zone. Some researchers have shown an increase in the size of the brain injury zone in the delayed period [9]. In our study, we estimated the size of the damage by MRI in each animal 24 h after the onset of ischemia and after 4 days. At the end of the experiment, the size of the affected area was compared using MRI and when staining the brain using TTC after decapitation of animals. During the analysis of MRI images, the area of intense color change was taken into account as the area of damage (the area of dead neurons was lighter in color than that of healthy brain tissue). Of course, it is more difficult to assess the affected area with MRI than with TTC brain staining, since the determination of the boundaries of the lesion by the intensity of staining depends on the researcher and is rather subjective. In addition, it is impossible to clearly delineate the zone of necrosis from the zone of ischemic penumbra.

As mentioned earlier, the mortality rate of animals in the vehicle group was very high. Most of the animals in the vehicle group did not survive to the end of the experiment and were not included in the final data calculation. However, the size of the brain damage zone in animals that survived 24 h but did not survive until the end of the experiment was comparable to the data obtained in the first series of experiments and amounted to 44.2 ± 10.2%, and the maximum damage in one animal was more than half of the ipsilateral hemisphere (55%). Thus, in the vehicle group, only animals with a small area of brain damage, which was comparable to brain lesion in animals receiving ubiquinol, survived to the end of the experiment. The size of the brain lesion in the vehicle group was 24.4%, and in the ubiquinol group 24.7%. Within four days, the necrosis zone in the vehicle group doubled and amounted to 48%. In the ubiquinol group, there was no increase in the affected area by day 4; the damage remained within 24.5%. Thus, it was shown that a single intravenous administration of ubiquinol in the acute period of ischemia not only reduced the lesion focus for 1 day due to the accumulation of coenzyme Q10 in the brain cells and the restoration of the antioxidant status of the brain tissue, but also prevented its further increase due to the decrease in pathogenetic cascades, underlying the delayed growth of the brain damage zone.

The key difficulty in the brain damage area analysis after ischemia using MRI images is the absence of a clearly visualized border of ischemic tissue. Thus, it is impossible to clearly visualize the area of necrosis, but it is the general area of injury that is assessed: necrosis plus the surrounding ischemic tissue. At the end of our experiment on day 4, we conducted a comparative assessment of the size of the brain lesion using MRI and brain staining using the metabolic dye TTC. With this staining, it is possible to clearly visualize the area of brain necrosis—it is a white tissue that does not stain, while TTC stains relatively viable brain tissue in a deep red color. In our study, it was shown that on day 4, when analyzing MRI images, the damage zone was visualized more clearly than on the first day (Figure 11).

In the vehicle group, brain lesion was 48%, and did not significantly differ from the size of the damaged zone when assessed using TTC staining (39%). In the ubiquinol group, the damage zone by MRI analysis was 24.5%, and in the assessment of TTC staining, it was 22.7%. Thus, it can be assumed that the zone of brain damage by the 4th day after ischemia is represented mainly by necrosis, which is clearly visualized by TTC staining and visualized on MRI images. This makes the analysis of MRI a reliable method for assessing the dynamics of the zone of necrosis in one animal over a long time.

Despite the registration of coenzyme Q10 in the rat brain using HPLC being a simple and widespread method, there are few studies describing CoQ10 change during ischemia [22]. In our study, when analyzing the tissue content of coenzyme Q10 in brain tissue, it was shown that ischemia leads to depletion of the endogenous coenzyme Q10 reserve in both hemispheres (Table 1).

A more severe decrease was noted in the ipsilateral hemisphere, in which the necrosis zone was located. However, the amount of endogenous CoQ10 was also depleted in the contralateral hemisphere, most likely because the brain is a common functioning system and pathological ischemic processes affect both hemispheres. In previous pharmacokinetic studies on an innovative dosage form of ubiquinol, it was shown that four days after a single intravenous administration, the level of coenzyme Q10 in the brain remains significantly increased [13]. Intravenous administration of ubiquinone and ubiquinol resulted in the restoration of total coenzyme Q10 level in the brain in both hemispheres to the value of native animals. Thus, it can be assumed that it was the persistent accumulation of coenzyme Q10 in the brain that made it possible to realize the neuroprotective effect, which was expressed in limiting the zone of necrosis and improving the neurological functions in animals. In the delayed period for 4 days, there was a further decrease in the level of coenzyme Q10 in the brain. At the same time, in the saline and vehicle groups, the ipsilateral hemisphere was the most affected—it showed a significant decrease in the level of CoQ10 both in comparison with the contralateral hemisphere and with native animal brain. In the ubiquinol group, a tendency to decrease in the tissue content of CoQ10 in the brain was noted, but this decrease was not statistically significant compared with the native animals. Thus, a single intravenous administration of ubiquinol provided a persistent increase in the level of coenzyme Q10 in the brain for at least four days.

## 5. Conclusions

Thus, in this study, we have shown the strong neuroprotective efficacy of a new innovative form of ubiquinol for intravenous administration. Intravenous administration of ubiquinol after the onset of cerebral ischemia led to persistent accumulation of coenzyme Q10 in the rat brain, decreased mortality, improved neurological outcomes, and limited the size of brain necrosis both in the acute period of ischemia and in the delayed period for at least four days. The data obtained may indicate that intravenous administration of ubiquinol may be effective in the treatment of stroke as an additional neuroprotective drug, when administered immediately before thrombectomy or thrombolytic therapy.

## Figures and Tables

**Figure 1 antioxidants-09-01240-f001:**
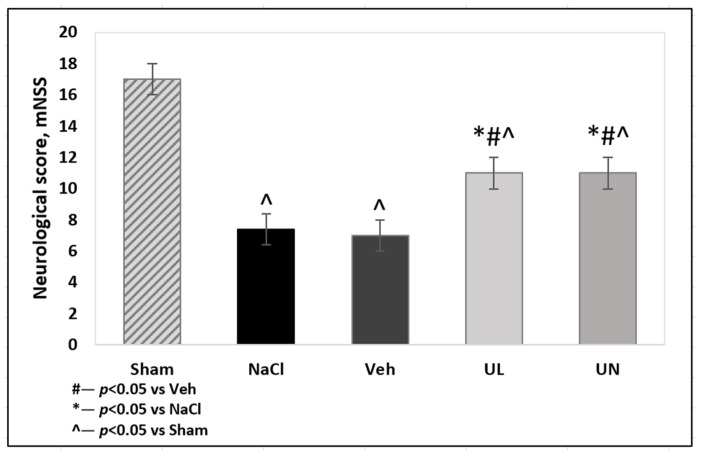
Neurological score, measured by mNSS scale 24 h after middle cerebral artery occlusion. Groups: Veh–MCAO+Vehicle; NaCL–MCAO+NaCL; UL–MCAO+Ubiquinol; UN–MCAO+Ubiquinone; Sham-operated.

**Figure 2 antioxidants-09-01240-f002:**
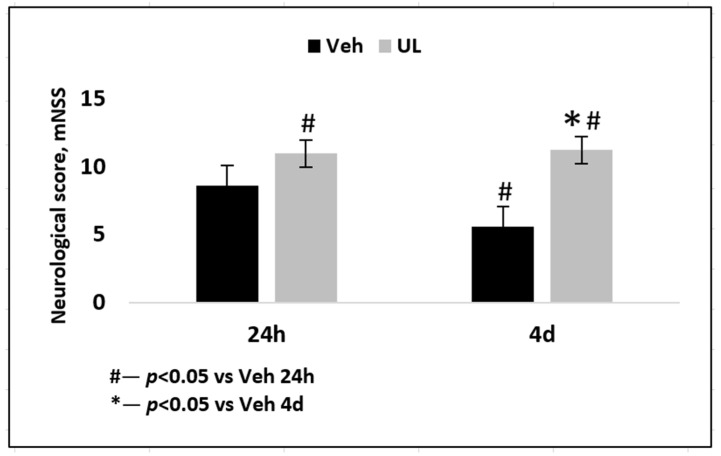
Neurological score, measured by mNSS scale 24 h and 4 days after middle cerebral artery occlusion. Groups: Veh–MCAO+Vehicle; UL–MCAO+Ubiquinol.

**Figure 3 antioxidants-09-01240-f003:**
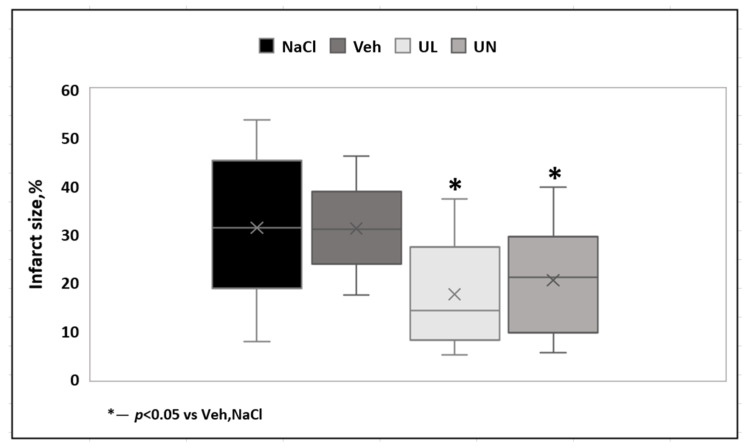
Brain infarct size (% to ipsilatheral hemisphere) 24 hours after middle cerebral artery occlusion. Groups: NaCL–MCAO+NaCL; Veh–MCAO+Vehicle; UL–MCAO+Ubiquinol; UN–MCAO+Ubiquinone.

**Figure 4 antioxidants-09-01240-f004:**
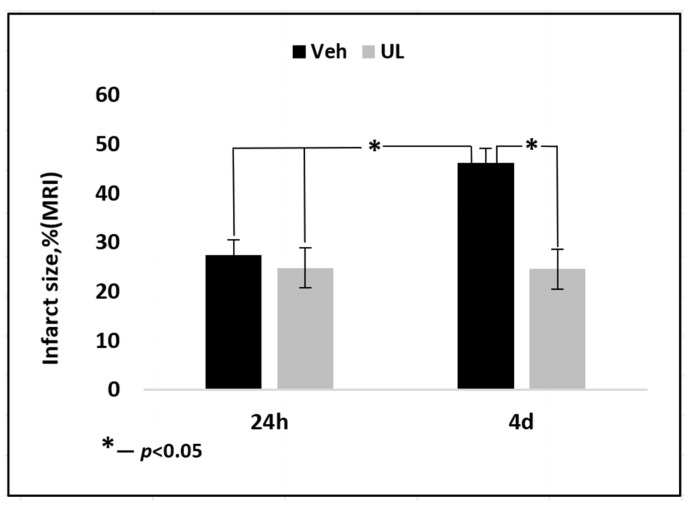
Brain infarct size (% to ipsilatheral hemisphere) 24 h and 4 days after middle cerebral artery occlusion. Groups: Veh–MCAO+Vehicle; UL–MCAO+Ubiquinol.

**Figure 5 antioxidants-09-01240-f005:**
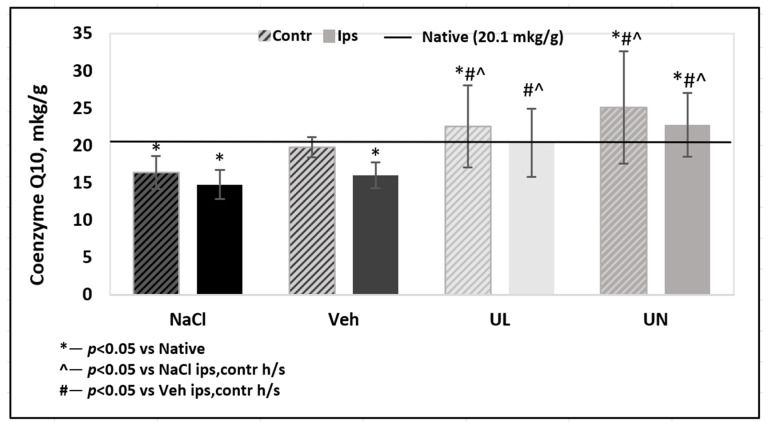
Coenzyme Q10 brain level measured by HPLC 24 hours after middle cerebral artery occlusion. Groups: NaCL–MCAO+NaCL; Veh–MCAO+Vehicle; UL–MCAO+Ubiquinol; UN–MCAO+Ubiquinone. Contr—contrlatheral hemisphere; Ips—ipsilatheral hemisphere. Baseline—CoQ10 level in native group.

**Figure 6 antioxidants-09-01240-f006:**
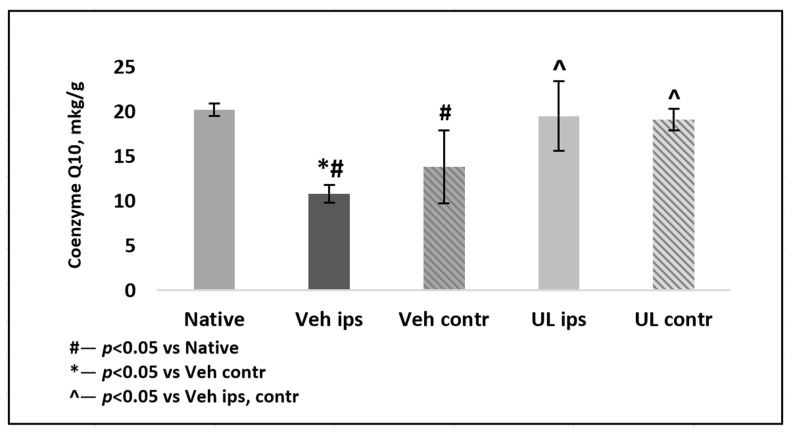
Coenzyme Q10 brain level measured by HPLC four days after middle cerebral artery occlusion. Groups: Veh–MCAO+Vehicle; UL–MCAO+Ubiquinol; Native. Contr—contrlatheral hemisphere; Ips—ipsilatheral hemisphere.

**Figure 7 antioxidants-09-01240-f007:**
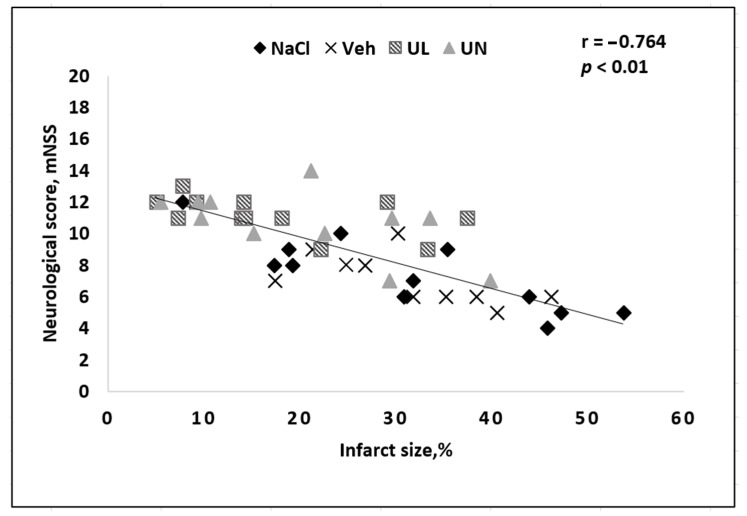
Correlation between the size of the brain infarct size and the severity of neurological deficit in rats after occlusion of the middle cerebral artery. Groups: NaCl–MCAO+NaCL; Veh–MCAO+Vehicle; UL–MCAO+Ubiquinol; UN–MCAO+Ubiquinone.

**Figure 8 antioxidants-09-01240-f008:**
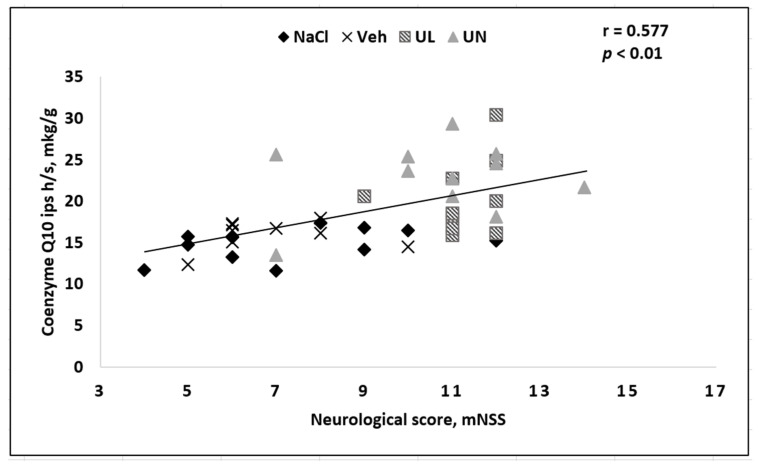
Correlation between the severity of neurological deficit and CoQ10 brain level in the ipsilateral hemisphere in rats after occlusion of the middle cerebral artery. Groups: NaCl–MCAO+NaCL; Veh–MCAO+Vehicle; UL–MCAO+Ubiquinol; UN–MCAO+Ubiquinone.

**Figure 9 antioxidants-09-01240-f009:**
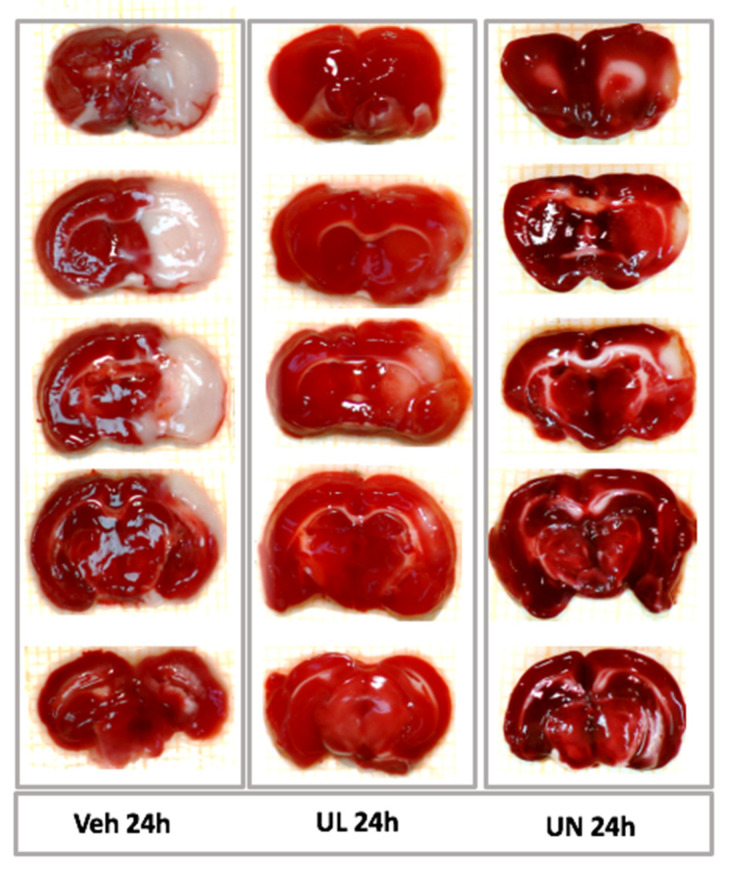
Sections of the brain 24 h after the middle cerebral artery occlusion. TTC staining. Viable tissue is stained with crimson, the zone of brain necrosis is white. Groups: Veh 24 h–MCAO+Vehicle; UL 24 h–MACO+Ubiquinol; UN 24 h–MCAO+Ubiquinone.

**Figure 10 antioxidants-09-01240-f010:**
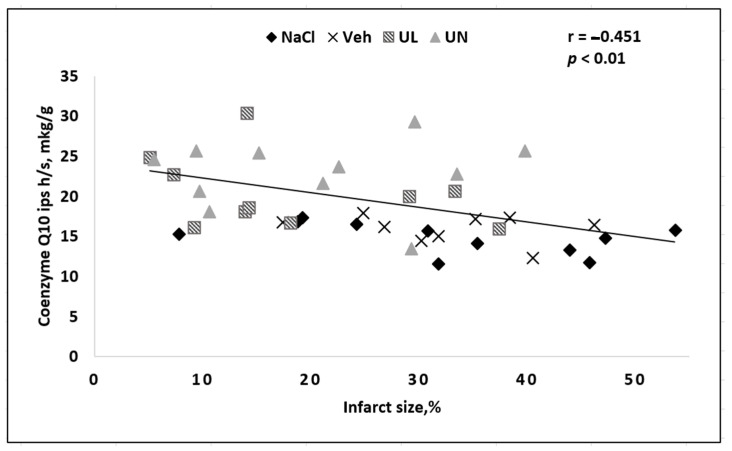
Correlation between the brain infarct size and CoQ10 brain level in the ipsilateral hemisphere in rats after occlusion of the middle cerebral artery. Groups: NaCl–MCAO+NaCL; Veh–MCAO+Vehicle; UL–MCAO+Ubiquinol; UN–MCAO+Ubiquinone.

**Figure 11 antioxidants-09-01240-f011:**
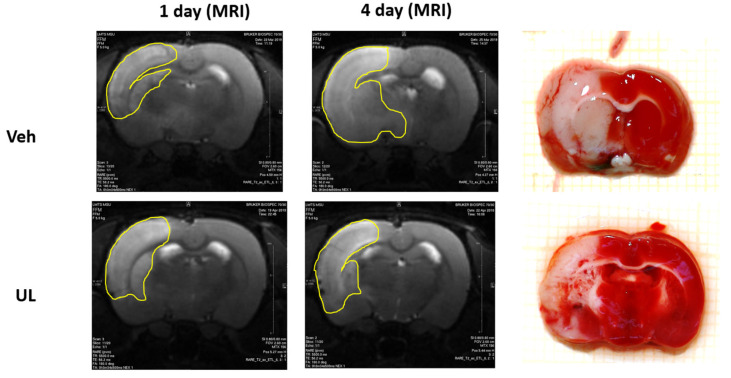
Dynamics of the brain lesion over four days and a comparison by MRI and TTC-staining. Veh–MCAO+Vehicle; UL–MCAO+Ubiquinol. The yellow line shows the area of brain damage.

**Table 1 antioxidants-09-01240-t001:** Coenzyme Q10 brain level 24 h and 4 days after middle cerebral artery occlusion. Groups: Native; Veh–MCAO+Vehicle; UL–MCAO+Ubiquinol. Ipsi h/s—ipsilateral hemisphere; Contr h/s—contralateral hemisphere. * *p* < 0.05 vs. native; ^#^
*p* < 0.05 vs. Veh contr h/s; ^&^
*p* < 0.05 vs. Veh ipsi h/s.

	Native	Veh		UL	
		Ipsi h/s	Contr h/s	Ipsi h/s	Contr h/s
**24 h**	20.2 ± 1.7	16.0 ± 1.7 *	19.7 ± 1.4 ^&^	20.4 ± 4.4 ^&^	22.6 ± 5.5 ^&^
**4 d**	20.2 ± 1.7	10.8 ± 0.7 *^#^	13.8 ± 1.0 *	19.5 ± 4.1 ^&^	19.1 ± 3.9 ^&^

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
