# Peer review of "Intravenous Administration of Coenzyme Q10 in Acute Period of Cerebral Ischemia Decreases Mortality by Reducing Brain Necrosis and Limiting Its Increase within 4 Days in Rat Stroke Model"

_antioxidants, 2020, doi:10.3390/antiox9121240_

Round 1

Reviewer 1 Report

Comments and Suggestions for Authors

This is a highly interesting paper reporting results from a preclinical study using a novel formulation of Coenzyme Q10 in ischemic stroke. The study has been well designed and investigates the therapeutic action by numerous methods. Although there are some minor design limitations (please see below), the results are meaningful and relevant.

I also think that the study is very transparent as realistic death rates after MCAO were reported, along with other methodological failures if they had occurred. The authors also provide a comprehensive, mainly well-balanced discussion.

Nevertheless, there is some room for improvement. Please find below my suggestions and recommendations.

Major points

(1) Please provide reasons for differences in animal numbers per group. Where there drop-outs or inclusion/exclusion criteria. The differences might not be explained by the drop-outs due to post-MCAO death as described in 2.1. What was the original number of animals enrolled? What means “intact animals”, may be “naïve” (not even sham-operated)?

(2) Please explain how infarct induction was controlled for. Please also explain in how brain damage was assessed.

(3) A NaCl group is reported in Figures 1, 3, and 5 but nowhere else. Please clarify and explain.

(4) Please avoid any speculation, in particular in results. There is no data on the penumbra so the assumption provided in lines 291-293 is purely speculative.

(5) A drawback is the short post-stroke surveillance period, which is much shorter than recommendations for the field (PMID 19246690). A rationale for the chosen approach must be given, and the need for future long-term observation studies of 28 days or more should be discussed.

(6) The application of the new therapeutic agent might be discussed in light of recent recommendations of using neuroprotective approaches in the era of thrombectomy (PMID 31166683, 27586682) and suggesting the evaluation of adjunctive neuroprotective interventions – just as done here. Thus, the reported substance may be beneficial in that field as well, as impressively suggested by the experiments.

 Minor points

(7) Can you please clarify whether affiliation 3 (House of Pharmacy) refers to a private/industrial or public institution? Please also clarify whether it is a for-profit or non-profit institution – just for the sake of completeness.

(8) Although generally written well, there a number typos, for instance:

  • ”…rats mortality rate…” (line 19); shall read: “…rats’…”
  • “…pathogenetic […] symptomatic action…” (line 37); this is misleading. It is unclear what “pathgenetic means” and successful symptomatic therapy in stroke would already be very good. May be just “…search for new therapeutic agents…”?
  • “… we perform two independent experiments.” (line 80); shall read: “performed”
  • “Reglodi D” (line 146); shall read: “Reglodi et al.”
  • “17.7±1,3” (line 217); shall read “1.3”; in general use full stops, no commas for decimal numbers (e.g., lines 247, 251, 273)
  • “…this decrease was not significant compare with…” (line 335); shall read “….as compared to the…”
  • There are occasional redundant blank spaces for instance in line 147 and 150 (between sentences). Please remove all of those redundant blank spaces.

Please also avoid the use of any synonyms, for instance “brain necrosis area” versus “brain damage”. Some phrases also seem to be a bit awkward (e.g., “Such a large number 48 of application points...”, “intravenous medicine administration”). This is not a major problem at this stage as authors are non-native speakers, but language correction by a native speaker or a professional language correction service with experience in medical writing is recommended. Language must be acceptable for acceptance of the manuscript.

(9) Please make sure that ALL introduction are properly introduced, for instance “mNSS”.

(10) In the introduction, thrombectomy should be mentioned as a treatment option for stroke as well.

(11) The mNSS is a rather imprecise test (PMID 28931616), what might be discussed as a potential limitation. More sensitive might have reported even more differences.

(12) Please indicate whether error indicators represent SEM or SD. SD is the appropriate error indicator (PMID 23881207), please correct in case SEM was used.

(13) Please provide individual data points for all experiments, this can be done as overlay to “dynamite bars” as they currently stand. Moreover, the different shades of grey and patterns in some diagrams do not transport any information and might be omitted. The use od symbols in correlation diagrams might be improved as well. Consider the use of colored dots rather than differently shaped symbols with or without pattern filling etc.

Please also be consistent in diagram style, for instance whether statistical significance was indicated with bars, or by using different significance indicator symbols. The resolution of the figures is generally poor and auxiliary lines in diagrams may not be needed.

(14) Please give exact p-values if P>0.05. Background: the null hypothesis can only be accepted at p>0.5 so 0.05≤p<0.4 is a range in which no definite conclusion can be made.

(15) The reported rodent data are meaningful and relevant. However, what is know on positive effects of the substance on the white matter? This might be best investigated in large animal stroke models, currently emerging (PMID 30732549). Those are also very feasible for simulating thrombectomy. It would be beyond the scope of this study to show such data, but authors may wish to discuss this potential future research option.

(16) Figure 9 and Figure 11 should be merged with respective diagrams.

(17) Figure 7 is rather interesting, but shows rather obvious findings that have been reported before. This is not needed in the main document, but a supplementary figure might be considered in case authors really wish to show that data.

Please note: “PMID” refers to PubMed Identification number for articles found at www.pubmed.gov

Author Response

Thank you for such a careful and detailed analysis of our article. We tried to take into account all the comments and make all the correctives. Below are the answers point by point:

  1. Thanks for your comment, animal amount have been corrected to their total number including mortality. Based on the first series of experiments and our previous experience with the model of middle cerebral artery occlusion, we assumed a mortality rate of about 20% in the vehicle group. Since in the second series of experiments we monitored the dynamics of the brain lesion in each rat for 4 days, the comparison of brain lesion was carried out within one animal, that is, each rat served as a control for itself. This made it possible to minimize the number of animals in each group to 5 in order to comply with ethical standards for minimizing the number of animals in experiments. Therefore, for the second experiment, 25 rats were selected in the vehicle group and 5 in the group with intravenous administration of ubiquinol.

We called intact a group of animals that did not undergo any manipulations, in contrast to sham-operated animals that underwent surgical manipulations. Since even minimal intervention could theoretically affect various physicochemical parameters, for example, the antioxidant status of the brain, we considered it expedient to add this group to the experiment. Thanks for your comment, we have changed the group name from intact to native.

  1. The model of intraluminal occlusion of middle cerebral artery is the gold standard in cerebral ischemia modeling in rodents and does not require additional control methods. In some studies, additional control of cerebral blood flow by Doppler is used, but this is not mandatory, requires craniotomy and can introduce additional trauma to the brain in the rat. In our experiment, we used Doccol single-use filaments, which were specially designed to perform this technique middle cerebral artery occlusion. The use of rats of the same weight and these standard filaments allowed us to obtain well reproducible results without additional control over the induction of cerebral ischemia.

Methods for measuring the area of brain damage are described in detail in Section 2.6. We used the ImageJ soft to evaluate both the TTC-stained sections and to analyze the MRI images. The damage zone was calculated according to the formula described in paragraph 2.6. The calculations were carried out by a blinded researcher.

  1. NaCL group was used only in the first experiment, which is presented in the figures with the data for the experiment for 24 hours (figure 1,3,5). Since in the first experiment we did not find a difference between the NaCL and vehicle groups, for the second experiment with the aim of minimizing the number of animals and adhering to ethical standards, it was decided to leave only the vehicle group as a comparison (as described in the discussion of lines 411-414). Figures 2,4,6 show comparisons of the data of the vehicle and ubiquinol on the 1st and 4th days of the experiment, therefore the NaCL group is not presented in these diagrams.

  1. Thank you for such a detailed study of our article. On your recommendation, we removed this statement, since it was our guess and did not have an exact evidence base.

  1. For experimental studies on modeling stroke in rodents, the period of 24 hours is widely used to assess the neuroprotective efficacy of substances in the acute period of cerebral ischemia. For most authors, the duration of the experiment of 24 hours is included in the protocols for studying the effectiveness of their substances. The duration of the second experiment of 4 days was chosen for 2 reasons. First, in the delayed period, we observed an increasing mortality of animals and the maximum occurred on the 4th day after ischemia. Secondly, based on the pharmacokinetic data of ubiquinol, that we earlier obtained, it was shown that its levels remain significantly increased in the brain up to 4 days after a single intravenous administration (as indicated in the lines 557-559). Thus, we were interested to study the potential delayed neuroprotective effect after a single administration of ubiquinol.

  1. Of course, thrombectomy, along with thrombolytic therapy, is a promising method in the treatment of stroke, which can reduce mortality and improve neurological outcomes of stroke. We have chosen the technique of reversible cerebral ischemia precisely because of the possibility of reperfusion. Based on the functions of coenzyme Q10, we assume that it is particularly effective in treating reperfusion syndrome. Since we have chosen the introduction period by 15 minutes before reperfusion, it is theoretically possible inject coenzyme Q10 immediately before the thrombectomy or thrombolysis will be performed.

  1. The House of Pharmacy is a private commercial organization that develops and studies new drugs and performs their preclinical research. Among the authors of the article is an employee of the House of Pharmacy (Makarov V.G.), who kindly provided us ubiquinone and ubiquinol solutions throughout the experiment. However, all scientific experiments were carried out in the laboratory of the cardiovascular system of the medical faculty of Lomonosov Moscow State University, whose employees are not related to House of Pharmacy. Thus, the study was of purely scientific interest and there is no conflict of interest.

  1. Thank you very much for your detailed work with our article, we tried to correct all the mistakes, unfortunately, with constant work with the text, some of them can still be missed. Thank you for your corrections in the wording in the text of the article, we tried to make all the corrections that you indicated.

Regarding to the use of synonyms, we do not consider the wording of the «brain necrosis area» and «brain damage» as equivalent expressions. In our view, it is impossible to clearly distinguish the area of ​​necrosis with MRI (the zone of necrosis and the zone of ischemic penumbra is visualized), therefore, when talking about the analysis of an MRI image, we use the term «brain damage». When staining TTС, it is already possible to clearly identify the area of ​​dead necrotic tissue, therefore we use the term «brain necrosis».

  1. Thanks for the note, we re-checked all the abbreviations for correctness, hopefully we fixed everything. However, mNSS is «modified neurological severity score» and abbreviated to mNSS.

  1. Thank you, we have added thrombectomy to the introduction as one of the methods of stroke therapy, which can reduce mortality and improve neurological outcomes of patients. Unfortunately, this method is not so widely used in our country so far, which is probably why we overlooked it.

  1. mNSS is widely used to detect neurological deficits in rodents after cerebral ischemia. It is commonly used by international authors in their works (here are some articles published in 2020: PMID 32241216, 32375835, 32576025). These tests are particularly good at detecting motor deficits - especially the tail hanging test and the vertical grid climb test. The technique is quite sensitive, but also quite simple to learn, which reduces the risk of error and the degree of subjectivity in assessing the deficit. Of course, the neurological tests were performed by a blinded researcher.

  1. Section 2.8 describes the statistical methods that we used. All data are present using the standard deviation (SD) as a more indicative parameter.

  1. In our article, we used the following method of graffiti and color differentiation: darker shades correspond to groups without administration of drugs (NaCL and vehicle), lighter shades correspond to groups of ubiquinol and ubiquinone. Hatching indicates the contralateral hemisphere, the absence of hatching indicates the ipsilateral hemisphere. These designations are uniform and remain in all figures. In our opinion, possible color differentiation improves the visual perception of graphs. Additional lines (for example, in the fig 4) are used in order to minimize the number of additional characters and improve clarity. However, based on your comment, we removed some of the lines from the background of the grays, which do not carry information and could obstruct perception.

  1. Thank you for your comment, we have corrected the p values to their exact value. We also remember the recent tendencies in the scientific community to label the real p level without reference to significance.

  1. According to international standards for conducting preclinical studies, any new drug must be tested on at least two species of different animals. Rodents, namely rats, are one of the types of animals that are excellent for preclinical studies, about 80% of all studies are carried out on rodents. Unfortunately, our technical capabilities allow us to carry out experiments only on rats, but we will definitely think about the prospects of experimenting on a different species.

  1. Photographs of brain slices are shown as an addition to figures in graphs for greater clarity. They show the most typical representatives of the studied groups. Presenting both photographs of slices and data in the form of graphs is standard practice and is common in articles on similar topics. In our point of view, combining photographs and graphs in one figure will be unnecessarily overloaded with information and will make it difficult to analyze. Therefore, if this remark is not critical, we would like to leave both photographs and graphics.

  1. This figure shows the correlation between the size of the brain lesion and the neurological status of the animals obtained in our experiment. Such data are, albeit expected, but new, obtained for the first time for coenzyme Q10 administration in this series of experiments. The authors would like to demonstrate this correlation in a separate graph to focus readers attention on it.

Reviewer 2 Report

This distinguished group of investigators explored the effects of intravenous administration of CoQ10 in acute period of cerebral ischemia in rats. Authors documented a persistent potential neuroprotective effect supporting a possible therapeutic role of ubiquinol in acute ischemia conditions.

The paper is nicely written, concise and interesting. Experimental design is robust. No major issues.

Minor suggestions: Please consider updating references by including the following recent clinical paper for better clarification of potential translational value of their findings in animals (i.e. CoQ10 supplementation and atherosclerosis, heart failure, acute CV disease, diabetes). I would suggest drafting 8-10 lines paragraph discussing the possible extension of the observed results in animals in clinical practice.

Author Response

Thank you for the careful review of our article!

On your advice, we have added a few phrases to the introduction about the use of coenzyme Q10 in the treatment of other diseases (please see lines 52-58):

"Because of coenzyme Q10 multitarget action, it finds application in many areas of medicine. Thus, it is used in the complex therapy of heart failure, diabetes and atherosclerosis. There are clinical studies that have shown the effectiveness of coenzyme Q10 in such neurodegenerative diseases as Parkinson's disease, Alzheimer's disease and Huntington's chorea, and a number of other diseases. However, in these studies, coenzyme Q10 was used orally for a long time at a high dosage, which is due to the low bioavailability when taking orally and the lack of forms for parenteral use at the time of the study. In such an urgent case as a stroke, intravenous administration of drugs is necessary."

We also added a phrase about the potential benefits of using coenzyme Q10 in the complex treatment of stroke in humans in conclusion (please see lines 576-579):

"The data obtained may indicate that intravenous administration of ubiquinol may be effective in the treatment of stroke as an additional neuroprotective drug when administered immediately before thrombectomy or thrombolytic therapy."